# A Probabilistic Framework for Nonlinearities in Stochastic Neural Networks

**Qinliang Su**     **Xuejun Liao**     **Lawrence Carin**
Department of Electrical and Computer Engineering
Duke University, Durham, NC, USA
{qs15, xjliao, lcarin}@duke.edu

## Abstract

We present a probabilistic framework for nonlinearities, based on doubly truncated Gaussian distributions. By setting the truncation points appropriately, we are able to generate various types of nonlinearities within a unified framework, including sigmoid, tanh and ReLU, the most commonly used nonlinearities in neural networks. The framework readily integrates into existing stochastic neural networks (with hidden units characterized as random variables), allowing one for the first time to *learn* the nonlinearities alongside model weights in these networks. Extensive experiments demonstrate the performance improvements brought about by the proposed framework when integrated with the restricted Boltzmann machine (RBM), temporal RBM and the truncated Gaussian graphical model (TGGM).

## 1   Introduction

A typical neural network is composed of nonlinear units connected by linear weights, and such a network is known to have universal approximation ability under mild conditions about the nonlinearity used at each unit [1, 2]. In previous work, the choice of nonlinearity has commonly been taken as a part of network design rather than network learning, and the training algorithms for neural networks have been mostly concerned with learning the linear weights. However, it is becoming increasingly understood that the choice of nonlinearity plays an important role in model performance. For example, [3] showed advantages of rectified linear units (ReLU) over sigmoidal units in using the restricted Boltzmann machine (RBM) [4] to pre-train feedforward ReLU networks. It was further shown in [5] that rectified linear units (ReLU) outperform sigmoidal units in a generative network under the same undirected and bipartite structure as the RBM.

A number of recent works have reported benefits of learning nonlinear units along with the inter-unit weights. These methods are based on using parameterized nonlinear functions to activate each unit in a neural network, with the unit-dependent parameters incorporated into the data-driven training algorithms. In particular, [6] considered the adaptive piecewise linear (APL) unit defined by a mixture of hinge-shaped functions, and [7] used nonparametric Fourier basis expansion to construct the activation function of each unit. The maxout network [8] employs piecewise linear convex (PLC) units, where each PLC unit is obtained by max-pooling over multiple linear units. The PLC units were extended to $L_p$ units in [9] where the normalized $L_p$ norm of multiple linear units yields the output of an $L_p$ unit. All these methods have been developed for learning the deterministic characteristics of a unit, lacking a stochastic unit characterization. The deterministic nature limits these methods from being easily applied to stochastic neural networks (for which the hidden units are random variables, rather than being characterized by a deterministic function), such as Boltzmann machines [10], restricted Boltzmann machines [11], and sigmoid belief networks (SBNs) [12].

We propose a probabilistic framework to unify the sigmoid, hyperbolic tangent (tanh) and ReLU nonlinearities, most commonly used in neural networks. The proposed framework represents a

unit $h$ probabilistically as $p(h|z, \boldsymbol{\xi})$, where $z$ is the total net contribution that $h$ receives from other units, and $\boldsymbol{\xi}$ represents the learnable parameters. By taking the expectation of $h$, a deterministic characterization of the unit is obtained as $\mathbb{E}(h|z, \boldsymbol{\xi}) \triangleq \int h \, p(h|z, \boldsymbol{\xi}) dh$. We show that the sigmoid, tanh and ReLU are well approximated by $\mathbb{E}(h|z, \boldsymbol{\xi})$ under appropriate settings of $\boldsymbol{\xi}$. This is different from [13], in which nonlinearities were induced by the additive noises of different variances, making the model learning much more expensive and nonlinearity producing less flexible. Additionally, more-general nonlinearities may be constituted or *learned*, with these corresponding to distinct settings of $\boldsymbol{\xi}$. A neural unit represented by the proposed framework is named a *truncated Gaussian (TruG)* unit because the framework is built upon truncated Gaussian distributions. Because of the inherent stochasticity, TruG is particularly useful in constructing stochastic neural networks.

The TruG generalizes the probabilistic ReLU in [14, 5] to a family of stochastic nonlinearities, with which one can perform two tasks that could not be done previously: (*i*) One can interchangeably use one nonlinearity in place of another under the same network structure, as long as they are both in the TruG family; for example, the ReLU-based stochastic networks in [14, 5] can be extended to new networks based on probabilistic tanh or sigmoid nonlinearities, and the respective algorithms in [14, 5] can be employed to train the associated new models with little modification; (*ii*) Any stochastic network constructed with the TruG can learn the nonlinearity alongside the network weights, by maximizing the likelihood function of $\boldsymbol{\xi}$ given the training data. We can learn the nonlinearity at the unit level, with each TruG unit having its own parameters; or we can learn the nonlinearity at the model level, with the entire network sharing the same parameters for all its TruG units. The different choices entail only minor changes in the update equation of $\boldsymbol{\xi}$, as will be seen subsequently.

We integrate the TruG framework into three existing stochastic networks: the RBM, temporal RBM [15] and feedforward TGGM [14], leading to three new models referred to as TruG-RBM, temporal TruG-RBM and TruG-TGGM, respectively. These new models are evaluated against the original models in extensive experiments to assess the performance gains brought about by the TruG. To conserve space, all propositions in this paper are proven in the Supplementary Material.

## 2  TruG: A Probabilistic Framework for Nonlinearities in Neural Networks

For a unit $h$ that receives net contribution $z$ from other units, we propose to relate $h$ to $z$ through the following stochastic nonlinearity,

$$p(h|z, \boldsymbol{\xi}) = \frac{\mathcal{N}\left(h \,|\, z, \sigma^2\right) \mathbb{I}(\xi_1 \leq h \leq \xi_2)}{\int_{\xi_1}^{\xi_2} \mathcal{N}\left(h' \,|\, z, \sigma^2\right) dh'} \triangleq \mathcal{N}_{[\xi_1, \xi_2]}\left(h \,|\, z, \sigma^2\right), \qquad (1)$$

where $\mathbb{I}(\cdot)$ is an indicator function and $\mathcal{N}\left(\cdot \,|\, z, \sigma^2\right)$ is the probability density function (PDF) of a univariate Gaussian distribution with mean $z$ and variance $\sigma^2$; the shorthand notation $\mathcal{N}_{[\xi_1, \xi_2]}$ indicates the density $\mathcal{N}$ is truncated and renormalized such that it is nonzero only in the interval $[\xi_1, \xi_2]$; $\boldsymbol{\xi} \triangleq \{\xi_1, \xi_2\}$ contains the truncation points and $\sigma^2$ is fixed.

The units of a stochastic neural network fall into two categories: *visible units* and *hidden units* [4]. The network represents a joint distribution over both hidden and visible units and the hidden units are integrated out to yield the marginal distribution of visible units. With a hidden unit expressed in (1), the expectation of $h$ is given by

$$\mathbb{E}(h|z, \boldsymbol{\xi}) = z + \sigma \frac{\phi(\frac{\xi_1 - z}{\sigma}) - \phi(\frac{\xi_2 - z}{\sigma})}{\Phi(\frac{\xi_2 - z}{\sigma}) - \Phi(\frac{\xi_1 - z}{\sigma})}, \qquad (2)$$

where $\phi(\cdot)$ and $\Phi(\cdot)$ are, respectively, the PDF and cumulative distribution function (CDF) of the standard normal distribution [16]. As will become clear below, a weighted sum of these expected hidden units constitutes the net contribution received by each visible unit when the hidden units are marginalized out. Therefore $\mathbb{E}(h|z, \boldsymbol{\xi})$ acts as a nonlinear activation function to map the incoming contribution $h$ receives to the outgoing contribution $h$ sends out. The incoming contribution received by $h$ may be a random variable or a function of data such as $z = \mathbf{w}^T \mathbf{x} + b$; the former case is typically for unsupervised learning and the latter case for supervised learning with $\mathbf{x}$ being the predictors.

By setting the truncation points to different values, we are able to realize many different kinds of nonlinearities. We plot in Figure 1 three realizations of $\mathbb{E}(h|z, \boldsymbol{\xi})$ as a function of $z$, each with a particular setting of $\{\xi_1, \xi_2\}$ and $\sigma^2 = 0.2$ in all cases. The plots of ReLU, tanh and sigmoid are

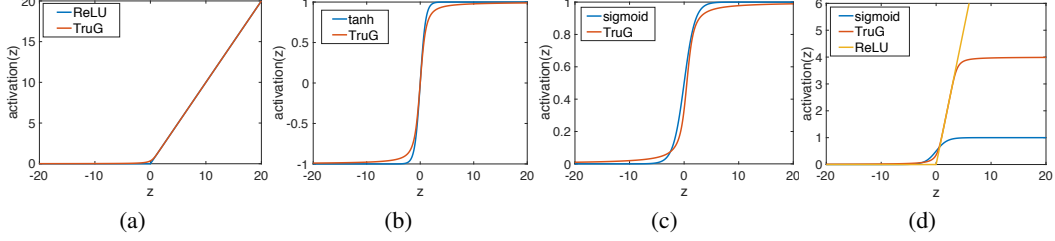

Figure 1: Illustration of different nonlinearities realized by the TruG with different truncation points. (a) $\xi_1 = 0$ and $\xi_2 = +\infty$; (b) $\xi_1 = -1$ and $\xi_2 = 1$; (c) $\xi_1 = 0$ and $\xi_2 = 1$; (d) $\xi_1 = 0$ and $\xi_2 = 4$.

also shown as a comparison. It is seen from Figure 1 that, by choosing appropriate truncation points, $\mathbb{E}(h|z, \boldsymbol{\xi})$ is able to approximate ReLU, tanh and sigmoid, the three types of nonlinearities most widely used in neural networks. We can also realize other types of nonlinearities by setting the truncation points to other values, as exemplified in Figure 1(d). The truncation points can be set manually by hand, selected by cross-validation, or learned in the same way as the inter-unit weights. In this paper, we focus on learning them alongside the weights based on training data.

The variance of $h$, given by [16],

$$\text{Var}(h|z, \boldsymbol{\xi}) = \sigma^2 + \sigma^2 \frac{\frac{\xi_1 - z}{\sigma}\phi\left(\frac{\xi_1 - z}{\sigma}\right) - \frac{\xi_2 - z}{\sigma}\phi\left(\frac{\xi_2 - z}{\sigma}\right)}{\Phi\left(\frac{\xi_2 - z}{\sigma}\right) - \Phi\left(\frac{\xi_1 - z}{\sigma}\right)} - \sigma^2 \left(\frac{\phi\left(\frac{\xi_1 - z}{\sigma}\right) - \phi\left(\frac{\xi_2 - z}{\sigma}\right)}{\Phi\left(\frac{\xi_2 - z}{\sigma}\right) - \Phi\left(\frac{\xi_1 - z}{\sigma}\right)}\right)^2, \tag{3}$$

is employed in learning the truncation points and network weights. Direct evaluation of (2) and (3) is prone to the numerical issue of $\frac{0}{0}$, because both $\phi(z)$ and $\Phi(z)$ are so close to 0 when $z < -38$ that they are beyond the maximal accuracy a double float number can represent. We solve this problem by using the fact that (2) and (3) can be equivalently expressed in terms of $\frac{\phi(z)}{\Phi(z)}$ by dividing both the numerator and the denominator by $\phi(\cdot)$. We make use of the following approximation for the ratio,

$$\frac{\phi(z)}{\Phi(z)} \approx \frac{\sqrt{z^2 + 4} - z}{2} \triangleq \gamma(z), \quad \text{for } z < -38, \tag{4}$$

the accuracy of which is established in Proposition 1.

**Proposition 1.** *The relative error is bounded by* $\left|\gamma(z)/\frac{\phi(z)}{\Phi(z)} - 1\right| < 2\frac{\sqrt{z^2 + 4} - z}{\sqrt{z^2 + 8} - 3z} - 1$; *moreover, for all* $z < -38$, *the relative error is guaranteed to be smaller than* $4.8 \times 10^{-7}$, *that is,* $\left|\gamma(z)/\frac{\phi(z)}{\Phi(z)} - 1\right| < 4.8 \times 10^{-7}$ *for all* $z < -38$.

## 3 RBM with TruG Nonlinearity

We generalize the ReLU-based RBM in [5] by using the TruG nonlinearity. The resulting TruG-RBM is defined by the following joint distribution over visible units $\mathbf{x}$ and hidden units $\mathbf{h}$,

$$p(\mathbf{x}, \mathbf{h}) = \frac{1}{Z} e^{-E(\mathbf{x}, \mathbf{h})} \mathbb{I}(\mathbf{x} \in \{0, 1\}^n, \xi_1 \leq \mathbf{h} \leq \xi_2), \tag{5}$$

where $E(\mathbf{x}, \mathbf{h}) \triangleq \frac{1}{2}\mathbf{h}^T \text{diag}(\mathbf{d})\mathbf{h} - \mathbf{x}^T \mathbf{W}\mathbf{h} - \mathbf{b}^T \mathbf{x} - \mathbf{c}^T \mathbf{h}$ is an energy function and $Z$ is the normalization constant. Proposition 2 shows (5) is a valid probability distribution.

**Proposition 2.** *The distribution* $p(\mathbf{x}, \mathbf{h})$ *defined in* (5) *is normalizable.*

By (5), the conditional distribution of $\mathbf{x}$ given $\mathbf{h}$ is still Bernoulli, $p(\mathbf{x}|\mathbf{h}) = \prod_{i=1}^n \sigma([\mathbf{W}\mathbf{h} + \mathbf{b}]_i)$, while the conditional $p(\mathbf{h}|\mathbf{x})$ is a truncated normal distribution, i.e.,

$$p(\mathbf{h}|\mathbf{x}) = \prod_{j=1}^m \mathcal{N}_{[\xi_1, \xi_2]}\left(h_j \middle| \frac{1}{d_j}[\mathbf{W}^T \mathbf{x} + \mathbf{c}]_j, \frac{1}{d_j}\right). \tag{6}$$

By setting $\xi_1$ and $\xi_2$ to different values, we are able to produce different nonlinearities in (6).

We train a TruG-RBM based on maximizing the log-likelihood function $\ell(\Theta, \boldsymbol{\xi}) \triangleq \sum_{\mathbf{x} \in \mathcal{X}} \ln p(\mathbf{x}; \Theta, \boldsymbol{\xi})$, where $\Theta \triangleq \{\mathbf{W}, \mathbf{b}, \mathbf{c}\}$ denotes the network weights, $p(\mathbf{x}; \Theta, \boldsymbol{\xi}) \triangleq \int_{\xi_1}^{\xi_2} p(\mathbf{x}, \mathbf{h}) d\mathbf{h}$ is contributed by a single data point $\mathbf{x}$, and $\mathcal{X}$ is the training dataset.

## 3.1 The Gradient w.r.t. Network Weights

The gradient w.r.t. $\Theta$ is known to be $\frac{\partial \ln p(\mathbf{x})}{\partial \Theta} = \mathbb{E}\left[\frac{\partial E(\mathbf{x}, \mathbf{h})}{\partial \Theta}\right] - \mathbb{E}\left[\frac{\partial E(\mathbf{x}, \mathbf{h})}{\partial \Theta}\Big| \mathbf{x}\right]$, where $\mathbb{E}[\cdot]$ and $\mathbb{E}[\cdot|\mathbf{x}]$ means the expectation w.r.t. $p(\mathbf{x}, \mathbf{h})$ and $p(\mathbf{h}|\mathbf{x})$, respectively. If we estimate the gradient using a standard sampling-based method, the variance associated with the estimate is usually very large. To reduce the variance, we follow the traditional RBM in applying the contrastive divergence (CD) to estimate the gradient [4]. Specifically, we approximate the gradient as

$$\frac{\partial \ln p(\mathbf{x})}{\partial \Theta} \approx \mathbb{E}\left[\frac{\partial E(\mathbf{x}, \mathbf{h})}{\partial \Theta}\Big| \mathbf{x}^{(k)}\right] - \mathbb{E}\left[\frac{\partial E(\mathbf{x}, \mathbf{h})}{\partial \Theta}\Big| \mathbf{x}\right], \tag{7}$$

where $\mathbf{x}^{(k)}$ is the $k$-th sample of the Gibbs sampler $p(\mathbf{h}^{(1)}|\mathbf{x}^{(0)}), p(\mathbf{x}^{(1)}|\mathbf{h}^{(1)}) \cdots p(\mathbf{x}^{(k)}|\mathbf{h}^{(k)})$, with $\mathbf{x}^{(0)}$ being the data $\mathbf{x}$. As shown in (6), $p(\mathbf{x}|\mathbf{h})$ and $p(\mathbf{h}|\mathbf{x})$ are factorized Bernoulli and univariate truncated normal distributions, for which efficient sampling algorithms exist [17, 18].

Furthermore, we can obtain that $\frac{\partial E(\mathbf{x}, \mathbf{h})}{\partial w_{ij}} = x_i h_j$, $\frac{\partial E(\mathbf{x}, \mathbf{h})}{\partial b_i} = x_i$, $\frac{\partial E(\mathbf{x}, \mathbf{h})}{\partial c_j} = h_j$ and $\frac{\partial E(\mathbf{x}, \mathbf{h})}{\partial d_j} = \frac{1}{2} h_j^2$. Thus estimation of the gradient with CD only requires $\mathbb{E}\left[h_j|\mathbf{x}^{(s)}\right]$ and $\mathbb{E}\left[h_j^2|\mathbf{x}^{(s)}\right]$, which can be calculated using (2) and (3). Using the estimated gradient, the weights can be updated using the stochastic gradient ascent algorithm or its variants.

## 3.2 The Gradient w.r.t. Truncation Points

The gradient w.r.t. $\xi_1$ and $\xi_2$ are given by

$$\frac{\partial \ln p(\mathbf{x})}{\partial \xi_1} = \sum_{j=1}^{m} \left(p(h_j = \xi_1) - p(h_j = \xi_1|\mathbf{x})\right), \tag{8}$$

$$\frac{\partial \ln p(\mathbf{x})}{\partial \xi_2} = \sum_{j=1}^{m} \left(p(h_j = \xi_2|\mathbf{x}) - p(h_j = \xi_2)\right), \tag{9}$$

for a single data point, with the derivation provided in the Supplementary Material. It is known that $p(h_j = \xi|\mathbf{x}) = \mathcal{N}_{[\xi_1, \xi_2]}\left(h_j = \xi \left| \frac{1}{d_j}[\mathbf{W}^T\mathbf{x} + \mathbf{c}]_j, \frac{1}{d_j}\right.\right)$, which can be easily calculated. However, if we calculate $p(h_j = \xi)$ directly, it would be computationally prohibitive. Fortunately, by noticing the identity $p(h_j = \xi) = \sum_{\mathbf{x}} p(h_j = \xi|\mathbf{x})p(\mathbf{x})$, we are able to estimate it efficiently with CD as $p(h_j = \xi) \approx p(h_j = \xi|\mathbf{x}^{(k)}) = \mathcal{N}_{[\xi_1, \xi_2]}\left(h_j = \xi \left| \frac{[\mathbf{W}^T\mathbf{x}^{(k)} + \mathbf{c}]_j}{d_j}, \frac{1}{d_j}\right.\right)$, where $\mathbf{x}^{(k)}$ is the $k$-th sample of the Gibbs sampler as described above. Therefore, the gradient w.r.t. the lower and upper truncation points can be estimated using the equations $\frac{\partial \ln p(\mathbf{x})}{\partial \xi_2} \approx \sum_{j=1}^{m}\left(p(h_j = \xi_2|\mathbf{x}) - p(h_j = \xi_2|\mathbf{x}^{(k)})\right)$ and $\frac{\partial \ln p(\mathbf{x})}{\partial \xi_1} \approx -\sum_{j=1}^{m}\left(p(h_j = \xi_1|\mathbf{x}) - p(h_j = \xi_1|\mathbf{x}^{(k)})\right)$. After obtaining the gradients, we can update the truncation points with stochastic gradient ascent methods.

It should be emphasized that in the derivation above, we assume a common truncation point pair $\{\xi_1, \xi_2\}$ shared among all units for the clarity of presentation. The extension to separate truncation points for different units is straightforward, by simply replacing (8) and (9) with $\frac{\partial \ln p(\mathbf{x})}{\partial \xi_{2j}} = (p(h_j = \xi_{2j}|\mathbf{x}) - p(h_j = \xi_{2j}))$ and $\frac{\partial \ln p(\mathbf{x})}{\partial \xi_{1j}} = (p(h_j = \xi_{1j}) - p(h_j = \xi_{1j}|\mathbf{x}))$, where $\xi_{1j}$ and $\xi_{2j}$ are the lower and upper truncation point of $j$-th unit, respectively. For the models discussed subsequently, one can similarly get the gradient w.r.t. unit-dependent truncations points.

After training, due to the conditional independence between $\mathbf{x}$ and $\mathbf{h}$ and the existence of efficient sampling algorithm for truncated normal, samples can be drawn efficiently from the TruG-RBM using the Gibbs sampler discussed below (7).

## 4 Temporal RBM with TruG Nonlinearity

We integrate the TruG framework into the temporal RBM (TRBM) [19] to learn the probabilistic nonlinearity in sequential-data modeling. The resulting temporal TruG-RBM is defined by

$$p(\mathbf{X}, \mathbf{H}) = p(\mathbf{x}_1, \mathbf{h}_1) \prod_{t=2}^{T} p(\mathbf{x}_t, \mathbf{h}_t | \mathbf{x}_{t-1}, \mathbf{h}_{t-1}), \tag{10}$$

where $p(\mathbf{x}_1, \mathbf{h}_1)$ and $p(\mathbf{x}_t, \mathbf{h}_t | \mathbf{x}_{t-1}, \mathbf{h}_{t-1})$ are both represented by TruG-RBMs; $\mathbf{x}_t \in \mathbb{R}^n$ and $\mathbf{h}_t \in \mathbb{R}^m$ are the visible and hidden variables at time step $t$, with $\mathbf{X} \triangleq [\mathbf{x}_1, \mathbf{x}_2, \cdots, \mathbf{x}_T]$ and $\mathbf{H} \triangleq [\mathbf{h}_1, \mathbf{h}_2, \cdots, \mathbf{h}_T]$. To be specific, the distribution $p(\mathbf{x}_t, \mathbf{h}_t | \mathbf{x}_{t-1}, \mathbf{h}_{t-1})$ is defined as $p(\mathbf{x}_t, \mathbf{h}_t | \mathbf{x}_{t-1}, \mathbf{h}_{t-1}) = \frac{1}{Z_t} \exp^{-E(\mathbf{x}_t, \mathbf{h}_t)} \mathbb{I}(\mathbf{x} \in \{0, 1\}^n, \xi_1 \le \mathbf{h}_t \le \xi_2)$, where the energy function takes the form $E(\mathbf{x}_t, \mathbf{h}_t) \triangleq \frac{1}{2} \Big( \mathbf{x}_t^T \text{diag}(\mathbf{a}) \mathbf{x}_t + \mathbf{h}_t^T \text{diag}(\mathbf{d}) \mathbf{h}_t - 2\mathbf{x}_t^T \mathbf{W}_1 \mathbf{h}_t - 2\mathbf{c}^T \mathbf{h}_t - 2(\mathbf{W}_2 \mathbf{x}_{t-1})^T \mathbf{h}_t - 2\mathbf{b}^T \mathbf{x}_t - 2(\mathbf{W}_3 \mathbf{x}_{t-1})^T \mathbf{x}_t - 2(\mathbf{W}_4 \mathbf{h}_{t-1})^T \mathbf{h}_t \Big)$; and $Z_t \triangleq \int_{-\infty}^{+\infty} \int_0^{+\infty} e^{-E(\mathbf{x}_t, \mathbf{h}_t)} d\mathbf{h}_t d\mathbf{x}_t$.

Similar to the TRBM, directly optimizing the log-likelihood is difficult. We instead optimize the lower bound

$$\mathcal{L} \triangleq \mathbb{E}_{q(\mathbf{H}|\mathbf{X})}[\ln p(\mathbf{X}, \mathbf{H}; \boldsymbol{\Theta}, \boldsymbol{\xi}) - \ln q(\mathbf{H}|\mathbf{X})], \tag{11}$$

where $q(\mathbf{H}|\mathbf{X})$ is an approximating posterior distribution of $\mathbf{H}$. The lower bound is equal to the log-likelihood when $q(\mathbf{H}|\mathbf{X})$ is exactly the true posterior $p(\mathbf{H}|\mathbf{X})$. We follow [19] to choose the following approximate posterior,

$$q(\mathbf{H}|\mathbf{X}) = p(\mathbf{h}_1|\mathbf{x}_1) \cdots p(\mathbf{h}_T|\mathbf{x}_{T-1}, \mathbf{h}_{T-1}, \mathbf{x}_T),$$

with which it can be shown that the gradient of the lower bound w.r.t. the network weights is given by $\frac{\partial \mathcal{L}}{\partial \boldsymbol{\Theta}} = \sum_{t=1}^{T} \mathbb{E}_{p(\mathbf{h}_{t-1}|\mathbf{x}_{t-2}, \mathbf{h}_{t-2}, \mathbf{x}_{t-1})} \Big( \mathbb{E}_{p(\mathbf{x}_t, \mathbf{h}_t|\mathbf{x}_{t-1}, \mathbf{h}_{t-1})} \Big[ \frac{\partial E(\mathbf{x}_t, \mathbf{h}_t)}{\partial \boldsymbol{\Theta}} \Big] - \mathbb{E}_{p(\mathbf{h}_t|\mathbf{x}_{t-1}, \mathbf{h}_{t-1}, \mathbf{x}_t)} \Big[ \frac{\partial E(\mathbf{x}_t, \mathbf{h}_t)}{\partial \boldsymbol{\Theta}} \Big] \Big)$. At any time step $t$, the outside expectation (which is over $\mathbf{h}_{t-1}$) is approximated by sampling from $p(\mathbf{h}_{t-1}|\mathbf{x}_{t-2}, \mathbf{h}_{t-2}, \mathbf{x}_{t-1})$; given $\mathbf{h}_{t-1}$ and $\mathbf{x}_{t-1}$, one can represent $p(\mathbf{x}_t, \mathbf{h}_t|\mathbf{x}_{t-1}, \mathbf{h}_{t-1})$ as a TruG-RBM and therefore the two inside expectations can be computed in the same way as in Section 3. In particular, the variables in $\mathbf{h}_t$ are conditionally independent given $(\mathbf{x}_{t-1}, \mathbf{h}_{t-1}, \mathbf{x}_t)$, i.e., $p(\mathbf{h}_t|\mathbf{x}_{t-1}, \mathbf{h}_{t-1}, \mathbf{x}_t) = \prod_{j=1}^{m} p(h_{jt}|\mathbf{x}_{t-1}, \mathbf{h}_{t-1}, \mathbf{x}_t)$ with each component equal to

$$p(h_{jt}|\mathbf{x}_{t-1}, \mathbf{h}_{t-1}, \mathbf{x}_t) = \mathcal{N}_{[\xi_1, \xi_2]} \left( h_{jt} \left| \frac{[\mathbf{W}_1^T \mathbf{x}_t + \mathbf{W}_2 \mathbf{x}_{t-1} + \mathbf{W}_4 \mathbf{h}_{t-1} + \mathbf{c}]_j}{d_j}, \frac{1}{d_j} \right. \right). \tag{12}$$

Similarly, the variables in $\mathbf{x}_t$ are conditionally independent given $(\mathbf{x}_{t-1}, \mathbf{h}_{t-1}, \mathbf{h}_t)$. As a result, $\mathbb{E}_{p(\mathbf{h}_t|\mathbf{x}_{t-1}, \mathbf{h}_{t-1}, \mathbf{x}_t)}[\cdot]$ can be calculated in closed-form using (2) and (3), and $\mathbb{E}_{p(\mathbf{x}_t, \mathbf{h}_t|\mathbf{x}_{t-1}, \mathbf{h}_{t-1}, \mathbf{x}_t)}[\cdot]$ can be estimated using the CD algorithm, as in Section Section 3. The gradient of $\mathcal{L}$ w.r.t. the upper truncation point is

$$\frac{\partial \mathcal{L}}{\partial \xi_2} = \mathbb{E}_{q(\mathbf{H}|\mathbf{X})} \left[ \sum_{t=1}^{T} \sum_{j=1}^{m} p(h_{jt} = \xi_2 | \mathbf{x}_{t-1}, \mathbf{h}_{t-1}, \mathbf{x}_t) - \sum_{t=1}^{T} \sum_{j=1}^{m} p(h_{jt} = \xi_2 | \mathbf{x}_{t-1}, \mathbf{h}_{t-1}) \right],$$

with $\frac{\partial \mathcal{L}}{\partial \xi_1}$ taking a similar form, where the expectations are similarly calculated using the same approach as described above for $\frac{\partial \mathcal{L}}{\partial \boldsymbol{\Theta}}$.

## 5 TGGM with TruG Nonlinearity

We generalize the feedforward TGGM model in [14] by replacing the probabilistic ReLU with the TruG. The resulting TruG-TGGM model is defined by the joint PDF over visible variables $\mathbf{y}$ and hidden variables $\mathbf{h}$,

$$p(\mathbf{y}, \mathbf{h}|\mathbf{x}) = \mathcal{N}(\mathbf{y}|\mathbf{W}_1 \mathbf{h} + \mathbf{b}_1, \sigma^2 \mathbf{I}) \mathcal{N}_{[\xi_1, \xi_2]}(\mathbf{h}|\mathbf{W}_0 \mathbf{x} + \mathbf{b}_0, \sigma^2 \mathbf{I}), \tag{13}$$

given the predictor variables $\mathbf{x}$. After marginalizing out $\mathbf{h}$, we get the expectation of $\mathbf{y}$ as

$$\mathbb{E}[\mathbf{y}|\mathbf{x}] = \mathbf{W}_1 \mathbb{E}(\mathbf{h}|\mathbf{W}_0\mathbf{x} + \mathbf{b}_0, \boldsymbol{\xi}) + \mathbf{b}_1, \tag{14}$$

where $\mathbb{E}(\mathbf{h}|\mathbf{W}_0\mathbf{x} + \mathbf{b}_0, \boldsymbol{\xi})$ is given element-wisely in (2). It is then clear that the expectation of $\mathbf{y}$ is related to $\mathbf{x}$ through the TruG nonlinearity. Thus $\mathbb{E}[\mathbf{y}|\mathbf{x}]$ yields the same output as a three-layer perceptron that uses (2) to activate its hidden units. Hence, the TruG-TGGM model defined in (13) can be understood as a stochastic perceptron with the TruG nonlinearity. By choosing different values for the truncation points, we are able to realize different kinds of nonlinearities, including ReLU, sigmoid and tanh.

To train the model by maximum likelihood estimation, we need to know the gradient of $\ln p(\mathbf{y}|\mathbf{x}) \triangleq \ln \int p(\mathbf{y}, \mathbf{h}|\mathbf{x}; \boldsymbol{\Theta}) d\mathbf{h}$, where $\boldsymbol{\Theta} \triangleq \{\mathbf{W}_1, \mathbf{W}_0, \mathbf{b}_1, \mathbf{b}_0\}$ represents the model parameters. By rewriting the joint PDF as $p(\mathbf{y}, \mathbf{h}|\mathbf{x}) \propto e^{-E(\mathbf{y}, \mathbf{h}, \mathbf{x})} I(\xi_1 \leq \mathbf{h} \leq \xi_2)$, the gradient is found to be given by $\frac{\partial \ln p(\mathbf{y}|\mathbf{x})}{\partial \boldsymbol{\Theta}} = \mathbb{E}\left[\frac{\partial E(\mathbf{y}, \mathbf{h}, \mathbf{x})}{\partial \boldsymbol{\Theta}}\Big|\mathbf{x}\right] - \mathbb{E}\left[\frac{\partial E(\mathbf{y}, \mathbf{h}, \mathbf{x})}{\partial \boldsymbol{\Theta}}\Big|\mathbf{x}, \mathbf{y}\right]$, where $E(\mathbf{y}, \mathbf{h}, \mathbf{x}) \triangleq \frac{||\mathbf{y}-\mathbf{W}_1\mathbf{h}-\mathbf{b}_1||^2 + ||\mathbf{h}-\mathbf{W}_0\mathbf{x}-\mathbf{b}_0||^2}{2\sigma^2}$; $\mathbb{E}[\cdot|\mathbf{x}]$ is the expectation w.r.t. $p(\mathbf{y}, \mathbf{h}|\mathbf{x})$; and $\mathbb{E}[\cdot|\mathbf{x}, \mathbf{y}]$ is the expectation w.r.t. $p(\mathbf{h}|\mathbf{x}, \mathbf{y})$. From (13), we know $p(\mathbf{h}|\mathbf{x}) = \mathcal{N}_{[\xi_1, \xi_2]}(\mathbf{h}|\mathbf{W}_0\mathbf{x} + \mathbf{b}_0, \sigma^2\mathbf{I})$ can be factorized into a product of univariate truncated Gaussian PDFs. Thus the expectation $\mathbb{E}[\mathbf{h}|\mathbf{x}]$ can be computed using (2). However, the expectations $\mathbb{E}[\mathbf{h}|\mathbf{x}, \mathbf{y}]$ and $\mathbb{E}[\mathbf{h}\mathbf{h}^T|\mathbf{x}, \mathbf{y}]$ involve a multivariate truncated Gaussian PDF and are expensive to calculate directly. Hence mean-field variational Bayesian analysis is used to compute the approximate expectations. The details are similar to those in [14] except that (2) and (3) are used to calculate the expectation and variance of $h$.

The gradients of the log-likelihood w.r.t. the truncation points $\xi_1$ and $\xi_2$ are given by $\frac{\partial \ln p(\mathbf{y}|\mathbf{x})}{\partial \xi_2} = \sum_{j=1}^{K} (p(h_j = \xi_2|\mathbf{y}, \mathbf{x}) - p(h_j = \xi_2|\mathbf{x}))$ and $\frac{\partial \ln p(\mathbf{y}|\mathbf{x})}{\partial \xi_1} = -\sum_{j=1}^{K} (p(h_j = \xi_1|\mathbf{y}, \mathbf{x}) - p(h_j = \xi_1|\mathbf{x}))$ for a single data point, with the derivation provided in the Supplementary Material. The probability $p(h_j = \xi_1|\mathbf{x})$ can be computed directly since it is a univariate truncated Gaussian distribution. For $p(h_j = \xi_2|\mathbf{y}, \mathbf{x})$, we approximate it with the mean-field marginal distributions obtained above.

Although TruG-TGGM involves random variables, thanks to the existence of close-form expression for the expectation of univariate truncated normal, the testing is still very easy. Given a predictor $\hat{\mathbf{x}}$, the output can be simply predicted with the conditional expectation $\mathbb{E}[\mathbf{y}|\mathbf{x}]$ in (14).

## 6 Experimental Results

We evaluate the performance benefit brought about by the TruG framework when integrated into the RBM, temporal RBM and TGGM. In each of the three cases, the evaluation is based on comparing the original network to the associated new network with the TruG nonlinearity. For the TruG, we either manually set $\{\xi_1, \xi_2\}$ to particular values, or learn them automatically from data. We consider both the case of learning a common $\{\xi_1, \xi_2\}$ shared for all hidden units and the case of learning a separate $\{\xi_1, \xi_2\}$ for each hidden unit.

**Results of TruG-RBM** The binarized MNIST and Caltech101 Silhouettes are considered in this experiment. The MNIST contains 60,000 training and 10,000 testing images of hand-written digits, while Caltech101 Silhouettes includes 6364 training and 2307 testing images of objects' silhouettes. For both datasets, each image has $28 \times 28$ pixels [22]. Throughout this experiment, 500 hidden units are used. RMSprop is used to update the parameters, with the delay and mini-batch size set to 0.95 and

Table 1: Averaged test log-probability on MNIST. ($\star$) Results reported in [20]; ($\diamond$) Results reported in [21] using RMSprop as the optimizer.

| Model | Trun. Points | Ave. Log-prob | |
|---|---|---|---|
| | | MNIST | Caltech101 |
| TruG-RBM | [0, 1] | -97.3 | -127.9 |
| | [0, +∞) | -83.2 | -105.2 |
| | [-1, 1] | -124.5 | -141.5 |
| | c-Learn | -82.9 | -104.6 |
| | s-Learn | **-82.5** | **-104.3** |
| RBM | — | -86.3$^\star$ | -109.0$^\diamond$ |

100, respectively. The weight parameters are initialized with the Gaussian noise of zero mean and 0.01 variance, while the lower and upper truncation points at all units are initialized to 0 and 1, respectively. The learning rates for weight parameters are fixed to $10^{-4}$. Since truncations points influence the whole networks in a more fundamental way than weight parameters, it is observed that smaller learning rates are often preferred for them. To balance the convergence speed and

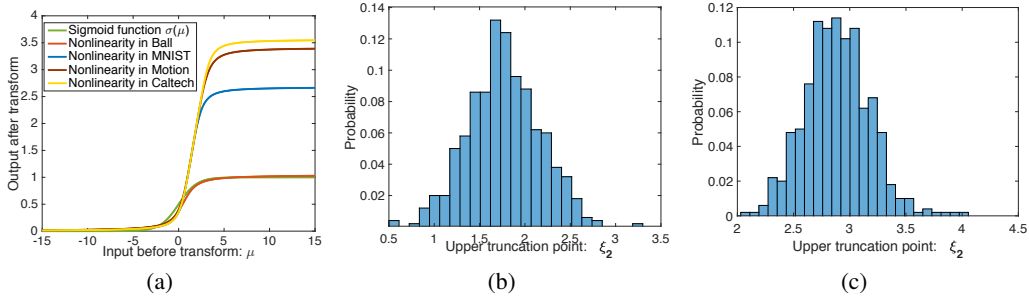

Figure 2: (a) The learned nonlinearities in TruG models with shared upper truncation point $\xi$; The distribution of unit-level upper truncation points of TruG-RBM for (b) MNIST; (c) Caltech101 Silhouettes.

performance, we anneal their learning rates from $10^{-4}$ to $10^{-6}$ gradually. The evaluation is based on the log-probability averaged over test data points, which are estimated using annealed importance sampling (AIS) [23] with $5 \times 10^5$ inverse temperatures equally spaced in $[0, 1]$; the reported test log-probability is averaged over 100 independent AIS runs.

To investigate the impact of truncation points, we first set the lower and upper truncation points to three fixed pairs: $[0, 1]$, $[0, +\infty)$ and $[-1, 1]$, which correspond to probabilistic approximations of sigmoid, ReLU and tanh nonlinearities, respectively. From Table 1, we see that the ReLU-type TruG-RBM performs much better than the other two types of TruG-RBM. We also learn the truncation points from data automatically. We can see that the model benefits significantly from nonlinearity learning, and the best performance is achieved when the units learn their own nonlinearities. The learned common nonlinearities (c-Learn) for different datasets are plotted in Figure 2(a), which shows that the model always tends to choose a nonlinearity in between sigmoid and ReLU functions. For the case with separate nonlinearities (s-Learn), the distributions of the upper truncation points in the TruG-RBM's for MNIST and Caltech101 Silhouettes are plotted in Figure 2(b) and (c), respectively. Note that due to the detrimental effect observed for negative truncation points, here the lower truncation points are fixed to zero and only the upper points are learned. To demonstrate the reliability of AIS estimate, the convergence plots of estimated log-probabilities are provided in Supplementary Material.

**Results of Temporal TruG-RBM** The Bouncing Ball and CMU Motion Capture datasets are considered in the experiment with temporal models. Bouncing Ball consists of synthetic binary videos of 3 bouncing balls in a box, with 4000 videos for training and 200 for testing, and each video has 100 frames of size $30 \times 30$. CMU Motion Capture is composed of data samples describing the joint angles associated with different motion types. We follow [24] to train a model on 31 sequences and test the model on two testing sequences (one is running and the other is walking). Both the original TRBM and the TruG-TRBM use 400 hidden units for Bouncing Ball and 300 hidden units for CMU Motion Capture. Stochastic gradient descent (SGD) is used to update the parameters, with the momentum set to 0.9. The learning rates are set to be $10^{-2}$ and $10^{-4}$ for the two datasets, respectively. The learning rate for truncation points is annealed gradually, as done in Section 6.

Since calculating the log-probabilities for these temporal models is computationally prohibitive, prediction error is employed here as the performance evaluation criteria, which is widely used [24, 25] in temporal generative models. The performances averaged over 20 independent runs are reported here. Tables 2 and 3 confirm again that models benefit remarkably from nonlinearity learning, especially in the case of learning a separate nonlinearity for each hidden unit. It is noticed that, although the ReLU-type TruG-TRBM performs better the tanh-type TruG-TRBM on Bouncing Ball, the former performs much worse than the latter on CMU Motion Capture. This demonstrates that a fixed nonlinearity cannot perform well on every dataset. However, by learning truncation points automatically, the TruG can adapt the nonlinearity to the data and thus performs the best on every dataset (up to the representational limit of the TruG framework). Video samples drawn from the trained models are provided in the Supplementary Material.

**Results of TruG-TGGM** Ten datasets from the UCI repository are used in this experiment. Following the procedures in [26], datasets are randomly partitioned into training and testing subsets for

Table 2: Test prediction error on Bouncing Ball. (⋆) Taken from [24], in which 2500 hidden units are used.

| Model | Trun. Points | Pred. Err. |
|---|---|---|
| TruG-TRBM | [0, 1] | 6.38±0.51 |
| | [0, +∞) | 4.16±0.42 |
| | [-1, 1] | 6.01±0.52 |
| | c-Learn | 3.82±0.41 |
| | s-Learn | **3.66±0.46** |
| TRBM | — | 4.90±0.47 |
| RTRBM⋆ | — | 4.00±0.35 |

Table 3: Test prediction error on CMU Motion Capture, in which 'w' and 'r' mean walking and running, respectively. (⋆) Taken from [24].

| Model | Trun. Points | Err. (w) | Err. (r) |
|---|---|---|---|
| TruG-TRBM | [0, 1] | 8.2±0.18 | 6.1±0.22 |
| | [0, +∞) | 21.8±0.31 | 14.9±0.29 |
| | [-1, 1] | 7.3±0.21 | 5.9±0.22 |
| | c-Learn | **6.7±0.29** | 5.5±0.22 |
| | s-Learn | 6.8±0.24 | **5.4±0.14** |
| TRBM | — | 9.6±0.15 | 6.8±0.12 |
| ss-SRTRBM⋆ | — | 8.1±0.06 | 5.9±0.05 |

Table 4: Averaged test RMSEs for multilayer perception (MLP) and TruG-TGGMs under different truncation points. (⋆) Results reported in [26], where BH, CS, EE, K8 NP, CPP, PS, WQR, YH, YPM are the abbreviations of Boston Housing, Concrete Strength, Kin8nm, Naval Propulsion, Cycle Power Plant, Protein Structure, Wine Quality Red, Yacht Hydrodynamic, Year Prediction MSD, respectively.

| Dataset | MLP (ReLU)⋆ | TruG-TGGM with Different Trun. Points | | | | |
|---|---|---|---|---|---|---|
| | | [0, 1] | [0, +∞) | [-1, 1] | c-Learn | s-Learn |
| BH | 3.228 ±0.195 | 3.564±0.655 | **3.214±0.555** | 4.003±0.520 | 3.401±0.375 | 3.622± 0.538 |
| CS | 5.977±0.093 | 5.210±0.514 | 5.106±0.573 | 4.977±0.482 | 4.910±0.467 | **4.743± 0.571** |
| EE | 1.098±0.074 | 1.168±0.130 | 1.252±0.123 | 1.069±0.166 | **0.881±0.079** | 0.913± 0.120 |
| K8 | 0.091±0.002 | 0.094±0.003 | 0.086±0.003 | 0.091±0.003 | **0.073±0.002** | 0.075± 0.002 |
| NP | **0.001±0.000** | 0.002±0.000 | 0.002±0.000 | 0.002± 0.000 | **0.001±0.000** | **0.001± 0.000** |
| CPP | 4.182±0.040 | 4.023±0.128 | 4.067±0.129 | 3.978±0.132 | 3.952±0.134 | **3.951± 0.130** |
| PS | 4.539±0.029 | 4.231±0.083 | 4.387±0.072 | 4.262±0.079 | 4.209±0.073 | **4.206± 0.071** |
| WQR | 0.645±0.010 | 0.662±0.052 | 0.644±0.048 | 0.659±0.052 | 0.645±0.050 | **0.643± 0.048** |
| YH | 1.182±0.165 | 0.871±0.367 | 0.821±0.276 | 0.846±0.310 | 0.803±0.292 | **0.793± 0.289** |
| YPM | 8.932±N/A | 8.961±N/A | 8.985±N/A | **8.859±N/A** | 8.893±N/A | 8.965± N/A |

10 trials except the largest one (Year Prediction MSD), for which only one partition is conducted due to computational complexity. Table 4 summarizes the root mean square error (RMSE) averaged over the different trials. Throughout the experiment, 100 hidden units are used for the two datasets (Protein Structure and Year Prediction MSD), while 50 units are used for the remaining. RMSprop is used to optimize the parameters, with RMSprop delay set to 0.9. The learning rate is chosen from the set $\{10^{-3}, 2 \times 10^4, 10^{-4}\}$, while the mini-batch size is set to 100 for the two largest datasets and 50 for the others. The number of VB cycles used in the inference is set to 10 for all datasets.

The RMSE's of TGGMs with fixed and learned truncation points are reported in Table 4, along with the RMSE's of the (deterministic) multilayer perceptron (MLP) using ReLU nonlinearity for comparison. Similar to what we have observed in generative models, the supervised models also benefit significantly from nonlinearity learning. The TruG-TGGM with learned truncation points perform the best for most datasets, with the separate learning performing slightly better than the common learning overall. Due to the limited space, the learned nonlinearities and their corresponding truncation points are provided in Supplementary Material.

## 7 Conclusions

We have presented a probabilistic framework, termed *TruG*, to unify ReLU, sigmoid and tanh, the most commonly used nonlinearities in neural networks. The TruG is a family of nonlinearities constructed with doubly truncated Gaussian distributions. The ReLU, sigmoid and tanh are three important members of the TruG family, and other members can be obtained easily by adjusting the lower and upper truncation points. A big advantage offered by the TruG is that the nonlinearity is learnable from data, alongside the model weights. Due to its stochastic nature, the TruG can be readily integrated into many stochastic neural networks for which hidden units are random variables. Extensive experiments have demonstrated significant performance gains that the TruG framework can bring about when it is integrated with the RBM, temporal RBM, or TGGM.

## Acknowledgements

The research reported here was supported by the DOE, NGA, NSF, ONR and by Accenture.

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
