[Supplementary Material · TruG_supplementary.pdf]

# Supplementary Materials for 'A Probabilistic Framework for Nonlinearities in Stochastic Neural Networks'

**Qinliang Su**      **Xuejun Liao**      **Lawrence Carin**
Department of Electrical and Computer Engineering
Duke University, Durham, NC, USA
{qs15, xjliao, lcarin}@duke.edu

## 1 Proof of Proposition 1

*Proof.* It has been proved in [1, 2] that the inequality $\frac{\sqrt{z^2+8}-3z}{4} < \frac{\phi(z)}{\Phi(z)} < \frac{\sqrt{z^2+4}-z}{2}$ always holds for $z < 0$. Thus, we can derive the bound $2\frac{\sqrt{z^2+4}-z}{\sqrt{z^2+8}-3z} - 1$ for the relative error by applying the lower and upper bounds. To prove the second part, we observe that the derivative of the upper bound with respect to (w.r.t.) $z$ is $\frac{2\left(z(z^2+4)^{-\frac{1}{2}}-1\right)\left((z^2+8)^{\frac{1}{2}}-3z\right)-\left((z^2+4)^{\frac{1}{2}}-z\right)\left(z(z^2+8)^{-\frac{1}{2}}-3\right)}{\left((z^2+8)^{\frac{1}{2}}-3z\right)^2}$. By using the relation $\frac{\sqrt{z^2+8}-3z}{4} < \frac{\sqrt{z^2+4}-z}{2}$ from the lower and upper bounds of $\frac{\phi(z)}{\Phi(z)}$, we can readily prove that the derivative is always positive for all $z < -38$. Therefore, we have that the upper bound for any $z < -38$ is smaller than the bound evaluated at $z = -38$, which is equal to $4.8 \times 10^{-7}$. $\qquad\square$

## 2 Proof of Proposition 2

*Proof.* We first derive $p^*(\mathbf{x}) \triangleq \int_{\xi_1}^{\xi_2} e^{-E(\mathbf{x},\mathbf{h})}d\mathbf{h}$. It can be readily obtained that $p^*(\mathbf{x}) = e^{\mathbf{b}^T\mathbf{x}} \prod_{j=1}^m \int_{\xi_1}^{\xi_2} e^{-\frac{1}{2}\left(d_j h_j^2 - 2[\mathbf{W}^T\mathbf{x}+\mathbf{c}]_j h_j\right)} dh_j$. Completing the square in the exponent of the integrand gives $p^*(\mathbf{x}) = e^{\mathbf{b}^T\mathbf{x}} \prod_{j=1}^m e^{\frac{[\mathbf{W}^T\mathbf{x}+\mathbf{c}]_j^2}{2d_j}} \cdot \int_{\xi_1}^{\xi_2} e^{-\frac{d_j}{2}\left(h_j - \frac{[\mathbf{W}^T\mathbf{x}+\mathbf{c}]_j}{d_j}\right)^2} dh_j$. By representing the integral with CDF functions, we obtain $p^*(\mathbf{x}) = e^{\mathbf{b}^T\mathbf{x}} \prod_{j=1}^m \frac{\Phi(\sqrt{d_j}\xi_2-\gamma_j) - \Phi(\sqrt{d_j}\xi_1-\gamma_j)}{\sqrt{d_j}\phi(\gamma_j)}$ with $\gamma_j \triangleq \frac{[\mathbf{W}^T\mathbf{x}+\mathbf{c}]_j}{\sqrt{d_j}}$. Since both $\Phi(\cdot)$ and $\phi(\cdot)$ can only be positive finite numbers, we observe that $p^*(\mathbf{x})$ is also finite for all $\mathbf{x} \in \{0,1\}^n$. Therefore, we can infer that the normalization constant $Z \triangleq \sum_{\mathbf{x}\in\{0,1\}^n} p^*(\mathbf{x})$ is finite. $\qquad\square$

## 3 Derivation for Gradient of Truncation Points in TruG-RBMs

It can be seen that $\frac{\partial \ln p(\mathbf{x};\Theta,\xi)}{\partial \xi_i} = \frac{\partial \ln s(\mathbf{x})}{\partial \xi_i} - \frac{\partial \ln Z}{\partial \xi_i}$, where $s(\mathbf{x}) \triangleq \int_{\xi_1}^{\xi_2} \cdots \int_{\xi_1}^{\xi_2} e^{-E(\mathbf{x},\mathbf{h};\Theta,\xi)} dh_1 \cdots dh_m$. We can easily see that $\frac{\partial \ln s(\mathbf{x})}{\partial \xi_2} = \frac{\frac{\partial}{\partial \xi_2} \int_{\xi_1}^{\xi_2} \cdots \int_{\xi_1}^{\xi_2} e^{-E(\mathbf{x},\mathbf{h};\Theta,\xi)} dh_1 \cdots dh_m}{\int_{\xi_1}^{\xi_2} \cdots \int_{\xi_1}^{\xi_2} e^{-E(\mathbf{x},\mathbf{h};\Theta,\xi)} dh_1 \cdots dh_m}$. According to Newton-Leibniz formula and multidimensional calculus, it can be shown that

$$\frac{\partial \ln s(\mathbf{x})}{\partial \xi_2} = \sum_{j=1}^m \int_{\xi_1}^{\xi_2} \frac{e^{-E(\mathbf{x},h_j=\xi_2,\mathbf{h}_{-j};\Theta,\xi)}}{\int_{\xi_1}^{\xi_2} e^{-E(\mathbf{x},\mathbf{h};\Theta,\xi)} d\mathbf{h}} d\mathbf{h}_{-j}, \tag{1}$$

where $\mathbf{h}_{-j}$ denotes the vector $\mathbf{h}$ without the $j$-th element; and for clarity, we abbreviate the multidimensional integral $\int \cdots \int f(\mathbf{h}) dh_1 \cdots dh_m$ as $\int f(\mathbf{h}) d\mathbf{h}$. By dividing the normalizer $Z$ for both the numerator and denominator, (1) becomes $\frac{\partial \ln s(\mathbf{x})}{\partial \xi_2} = \sum_{j=1}^{m} \int_{\xi_1}^{\xi_2} \frac{p(\mathbf{x}, h_j = \xi_2, \mathbf{h}_{-j})}{\int_{\xi_1}^{\xi_2} p(\mathbf{x}, \mathbf{h}; \Theta, \xi) d\mathbf{h}} d\mathbf{h}_{-j}$, which can be further written as

$$\frac{\partial \ln s(\mathbf{x})}{\partial \xi_2} = \sum_{j=1}^{m} p(h_j = \xi_2 | \mathbf{x}). \tag{2}$$

Following the similar procedures, it can be also derived that

$$\begin{aligned}
\frac{\partial \ln Z}{\partial \xi_2} &= \frac{\frac{\partial}{\partial \xi_2} \int_{\xi_1}^{\xi_2} \cdots \int_{\xi_1}^{\xi_2} \sum_{\mathbf{x}} e^{-E(\mathbf{x},\mathbf{h})} dh_1 \cdots dh_m}{\int_{\xi_1}^{\xi_2} \cdots \int_{\xi_1}^{\xi_2} \sum_{\mathbf{x}} e^{-E(\mathbf{x},\mathbf{h})} dh_1 \cdots dh_m} \\
&= \sum_{j=1}^{m} \sum_{\mathbf{x}} \int_{\xi_1}^{\xi_2} \frac{e^{-E(\mathbf{x}, h_j = \xi_2, \mathbf{h}_{-j})}}{\int_{\xi_1}^{\xi_2} \sum_{\mathbf{x}} e^{-E(\mathbf{x},\mathbf{h})} d\mathbf{h}} d\mathbf{h}_{-j} \\
&= \sum_{j=1}^{m} \sum_{\mathbf{x}} p(\mathbf{x}, h_j = \xi_2) \\
&= \sum_{j=1}^{m} p(h_j = \xi_2).
\end{aligned} \tag{3}$$

Thus, the gradient of $p(\mathbf{x}; \Theta, \boldsymbol{\xi})$ w.r.t. $\xi_2$ equals to

$$\frac{\partial \ln p(\mathbf{x}; \Theta, \boldsymbol{\xi})}{\partial \xi_2} = \sum_{j=1}^{m} \left( p(h_j = \xi_2 | \mathbf{x}) - p(h_j = \xi_2) \right); \tag{4}$$

With the similar derivations, it can be obtained that the gradient of $p(\mathbf{x}; \Theta, \boldsymbol{\xi})$ w.r.t. lower truncation point $\xi_1$ equals to

$$\frac{\partial \ln p(\mathbf{x}; \Theta, \boldsymbol{\xi})}{\partial \xi_1} = \sum_{j=1}^{m} \left( -p(h_j = \xi_1 | \mathbf{x}) + p(h_j = \xi_1) \right). \tag{5}$$

## 4   Partition Function Estimation for TruG-RBMs

To evaluate the model's performance, we need to calculate the partition function $Z$. By exploiting the bipartite structure in an RTGGM as well as the appealing properties of truncated normals, we show that we can use annealed importance sampling (AIS) [3, 4] to estimate it. Here, we only focus on the estimation for the RTGGM with binary data; methods for the other types data are derived similarly. By integrating out the hidden variables $\mathbf{h}$ in the joint pdf $p(\mathbf{x}, \mathbf{h}; \Theta) = \frac{1}{Z} e^{-\frac{1}{2} \left( \|\mathbf{D}^{\frac{1}{2}} \mathbf{h}\|^2 - 2\mathbf{x}^T \mathbf{W} \mathbf{h} - 2\mathbf{b}^T \mathbf{x} - 2\mathbf{c}^T \mathbf{h} \right)} \mathbb{I}(\mathbf{x} \in \{0,1\}^n, \xi_1 \leq \mathbf{h}_t \leq \xi_2)$, we obtain

$$p(\mathbf{x}; \Theta) = \frac{1}{Z} p^*(\mathbf{x}; \Theta) \mathbb{I}(\mathbf{x} \in \{0,1\}^n), \tag{6}$$

where $p^*(\mathbf{x}; \Theta)$ is defined in previous section.

Following the AIS procedure, we define two distributions $p_A(\mathbf{x}, \mathbf{h}^A) = \frac{1}{Z_A} e^{-E_A(\mathbf{x}, \mathbf{h}^A)} \mathbb{I}(\mathbf{x} \in \{0,1\}^n) \mathbb{I}(\xi_1 \leq \mathbf{h}_t \leq \xi_2)$ and $p_B(\mathbf{x}, \mathbf{h}^B) = \frac{1}{Z_B} e^{-E_B(\mathbf{x}, \mathbf{h}^B)} \mathbb{I}(\mathbf{x} \in \{0,1\}^n) \mathbb{I}(\xi_1 \leq \mathbf{h}_t \leq \xi_2)$, where $E_A(\mathbf{x}, \mathbf{h}^A) \triangleq \frac{1}{2}(\|\mathrm{diag}^{\frac{1}{2}}(\mathbf{d}) \mathbf{h}^A\|^2 - 2\mathbf{b}^{AT} \mathbf{x})$ and $E_B(\mathbf{x}, \mathbf{h}^B) \triangleq \frac{1}{2}(\|\mathrm{diag}^{\frac{1}{2}}(\mathbf{d}) \mathbf{h}^B\|^2 - 2\mathbf{x}^T \mathbf{W} \mathbf{h}^B - 2\mathbf{b}^T \mathbf{x} - 2\mathbf{c}^T \mathbf{h}^B)$. By construction, $p_0(\mathbf{x}, \mathbf{h}^A, \mathbf{h}^B) = p_A(\mathbf{x}, \mathbf{h}^A)$ and $p_K(\mathbf{x}, \mathbf{h}^A, \mathbf{h}^B) = p_B(\mathbf{x}, \mathbf{h}^B)$. The partition function of $p_A(\mathbf{x}, \mathbf{h}^A)$ can be obtained easily from (4) by simply setting $\mathbf{W}$ and $\mathbf{c}$ to be zero and then summing up all $\mathbf{x}$

$$Z_A = \prod_{i=1}^{n} (1 + e^{b_i^A}) \prod_{j=1}^{m} \frac{\Phi(\sqrt{d_j} \xi_2) - \Phi(\sqrt{d_j} \xi_1)}{\sqrt{d_j} \phi(0)}. \tag{7}$$

On the othe hand, the partition function of $p_B(\mathbf{x}, \mathbf{h}^B)$ can be approximated as

$$Z_B \approx \frac{\sum_{i=1}^{M} w^{(i)}}{M} Z_A, \tag{8}$$

where $w^{(i)}$ is constructed from a Markov chain that gradually transits from $p_A(\mathbf{x}, \mathbf{h}^A)$ to $p_B(\mathbf{x}, \mathbf{h}^B)$, with the transition realized via a sequence of intermediate distributions

$$p_k(\mathbf{x}, \mathbf{h}^A, \mathbf{h}^B) = \frac{1}{Z_k} e^{-(1-\beta_k)E_A(\mathbf{x}, \mathbf{h}^A) - \beta_k E_B(\mathbf{x}, \mathbf{h}^B)}$$
$$\times \mathbb{I}(\mathbf{x} \in \{0, 1\}^n, \xi_1 \le \mathbf{h}_t \le \xi_2), \tag{9}$$

where $0 = \beta_0 < \beta_1 < \ldots < \beta_K = 1$. By integrating out the latent variables $\mathbf{h}^A$ and $\mathbf{h}^B$, we can see that $p_k(\mathbf{x}) = \frac{1}{Z_k} p_k^*(\mathbf{x})$ with

$$p_k^*(\mathbf{x}) \triangleq e^{(1-\beta_k)\mathbf{b}^{AT}\mathbf{x}} e^{\beta_k \mathbf{b}^T \mathbf{x}}$$
$$\prod_{j=1}^{m} \frac{\Phi\left(\sqrt{\beta_k d_j}\xi_2 - \sqrt{\beta_k}\gamma_j\right) - \Phi\left(\sqrt{\beta_k d_j}\xi_1 - \sqrt{\beta_k}\gamma_j\right)}{\sqrt{\beta_k d_j}\phi\left(\sqrt{\beta_k}\gamma_j\right)}$$
$$\times \frac{\Phi\left(\sqrt{(1-\beta_k)d_j}\xi_2\right) - \Phi\left(\sqrt{(1-\beta_k)d_j}\xi_1\right)}{\sqrt{(1-\beta_k)d_j}\phi(0)}. \tag{10}$$

Based on the sequential pdfs $p_k(\mathbf{x}, \mathbf{h}^A, \mathbf{h}^B)$, the Markov chain $(\mathbf{x}_i^{(0)}, \mathbf{x}_i^{(1)}, \ldots, \mathbf{x}_i^{(K)})$ is simulated as $\mathbf{x}_i^{(0)} \sim p_0(\mathbf{x}_i, \mathbf{h}^A, \mathbf{h}^B)$, $(\mathbf{h}^A, \mathbf{h}^B) \sim p_1(\mathbf{h}^A, \mathbf{h}^B | \mathbf{x}_i^{(0)})$, $\mathbf{x}_i^{(1)} \sim p_1(\mathbf{x}_i | \mathbf{h}^A, \mathbf{h}^B)$, $\cdots$, $(\mathbf{h}^A, \mathbf{h}^B) \sim p_K(\mathbf{h}^A, \mathbf{h}^B | \mathbf{x}_i^{(K-1)})$ and $\mathbf{x}_i^{(K)} \sim p_K(\mathbf{x}_i | \mathbf{h}^A, \mathbf{h}^B)$. From the chain, a coefficient is constructed as $w^{(i)} = \frac{p_1^*(\tilde{\mathbf{x}}_i^{(0)})}{p_0^*(\tilde{\mathbf{x}}_i^{(0)})} \frac{p_2^*(\tilde{\mathbf{x}}_i^{(1)})}{p_1^*(\tilde{\mathbf{x}}_i^{(1)})} \cdots \frac{p_K^*(\tilde{\mathbf{x}}_i^{(K-1)})}{p_{K-1}^*(\tilde{\mathbf{x}}_i^{(K-1)})}$. Assuming $M$ independent Markov chains simulated in this way, one obtains $\{w^{(i)}\}_{i=1}^{M}$. Note that the Markov chains can be efficiently simulated, as all involved variables are conditionally independent.

## 5  Derivation for Gradient of Truncation Points in TruG-TGGMs

In order to learn the trunation points automatically, we need to derive the gradients $\frac{\partial \ln p(\mathbf{y}|\mathbf{x})}{\partial \xi_i} = \frac{\partial \ln s(\mathbf{y}|\mathbf{x})}{\partial \xi_i} - \frac{\partial \ln Z}{\partial \xi_i}$ for $i = 1, 2$, where $s(\mathbf{y}|\mathbf{x}) \triangleq \int_{\xi_1}^{\xi_2} e^{-E(\mathbf{y}, \mathbf{h}, \mathbf{x})} d\mathbf{h}$. Now, we can derive that

$$\frac{\partial \ln p(\mathbf{y}|\mathbf{x})}{\partial \xi_2} = \frac{\frac{\partial}{\partial \xi_2} \int_{\xi_1}^{\xi_2} \cdots \int_{\xi_1}^{\xi_2} e^{-E(\mathbf{y}, \mathbf{h}, \mathbf{x})} dh_1 \cdots dh_K}{\int_{\xi_1}^{\xi_2} \cdots \int_{\xi_1}^{\xi_2} e^{-E(\mathbf{y}, \mathbf{h}, \mathbf{x})} dh_1 \cdots dh_K}$$
$$= \sum_{j=1}^{K} \int_{\xi_1}^{\xi_2} \frac{e^{-E(\mathbf{y}, h_j = \xi_2, \mathbf{h}_{-j}, \mathbf{x})}}{\int_{\xi_1}^{\xi_2} e^{-E(\mathbf{y}, \mathbf{h}, \mathbf{x})} d\mathbf{h}} d\mathbf{h}_{-j}$$
$$= \sum_{j=1}^{K} p(h_j = \xi_2 | \mathbf{y}, \mathbf{x}), \tag{11}$$

and

$$\frac{\partial \ln Z}{\partial \xi_2} = \frac{\frac{\partial}{\partial \xi_2} \int_{\xi_1}^{\xi_2} \cdots \int_{\xi_1}^{\xi_2} e^{-E(\mathbf{x}, \mathbf{h})} dh_1 \cdots dh_K}{\int_{\xi_1}^{\xi_2} \cdots \int_{\xi_1}^{\xi_2} e^{-E(\mathbf{x}, \mathbf{h})} dh_1 \cdots dh_K}$$
$$= \sum_{j=1}^{K} \int_{\xi_1}^{\xi_2} \frac{e^{-E(\mathbf{x}, h_j = \xi_2, \mathbf{h}_{-j})}}{\int_{\xi_1}^{\xi_2} e^{-E(\mathbf{x}, \mathbf{h})} d\mathbf{h}} d\mathbf{h}_{-j}$$
$$= \sum_{j=1}^{K} p(h_j = \xi_2 | \mathbf{x}), \tag{12}$$

where $\mathbf{h}_{-j}$ means the vector $\mathbf{h}$ without the $j$-th element $h_j$. Similarly, it can be easily derived that $\frac{\partial \ln p(\mathbf{y}|\mathbf{x})}{\partial \xi_1} = -\sum_{j=1}^{K} p(h_j = \xi_1|\mathbf{y}, \mathbf{x})$ and $\frac{\partial \ln Z}{\partial \xi_1} = -\sum_{j=1}^{K} p(h_j = \xi_1|\mathbf{x})$. By combing the above expressions, we obtain

$$\frac{\partial \ln p(\mathbf{y}|\mathbf{x})}{\partial \xi_2} = \sum_{j=1}^{K} \left( p(h_j = \xi_2|\mathbf{y}, \mathbf{x}) - p(h_j = \xi_2|\mathbf{x}) \right), \tag{13}$$

$$\frac{\partial \ln p(\mathbf{y}|\mathbf{x})}{\partial \xi_1} = -\sum_{j=1}^{K} \left( p(h_j = \xi_1|\mathbf{y}, \mathbf{x}) - p(h_j = \xi_1|\mathbf{x}) \right). \tag{14}$$

The probability $p(h_j = \xi_i|\mathbf{x})$ can be computed directly since it is a univariate truncated normal distribution. For the term $p(h_j = \xi_2|\mathbf{y}, \mathbf{x})$, we will approximate it with the mean-field marginal distributions computed above.

## 6 Other Experimental Results

### 6.1 Convergence of AIS Estimation

To demonstrate the reliability of log-probabilities estimated using annealed importance sampling (AIS) algorithm [3] (reported in Table 1 in the paper), here we plot the estimate as a function of Gibbs sweeps. As seen in Figure 1, log-probabilities for both MNIST and Caltech101 Silhouettes datasets have already converged when we use $5 \times 10^5$ Gibbs sweeps.

Figure 1: Demonstration of convergence of estimated test log-probabilities. (a): MNIST; (b): Caltech101 Silhouettes.

### 6.2 Dynamics of Truncation Points

We plot the truncation point values as a function of time, so that we can examine how the truncation points evolves. We can see that for MNIST dataset, as the learning process proceed, the upper truncation point increases gradually, while for the Bouncing Ball dataset, the upper truncation point increases at the beginning and then decreases gradually.

### 6.3 Images Generated from TruG-RBM

Figures 3 show samples drawn from the TruG-RBM trained on MNIST and Caltech101 Silhouettes, respectively. As seen from the figure, the samples looks very similar to the true images and are also very diverse. This implies that the TruG-RBM has modeled these data very well.

### 6.4 Dictionaries Learned in Temporal TruG-RBM

Figures 4 shows the dictionaries learned in Temporal TruG-RBM.

### 6.5 Learned Nonlinearities and Truncation Points for TruG-TGGMs

The nonlinearities learned for the TruG-TGGM models on different datasets are plotted in Figure 5. Moreover, we also tabulate the learned truncation points on these datasets. Since we train the model 10 times for each dataset, the presented values in Table 1 are the average of the 10 trials.

Figure 2: The upper truncation point as a function of the number of training epochs, for (a) MNIST; (b) Bouncing Ball.

Figure 3: Samples drawn from TruG-RBM. (a): Trained on MNIST; (b): Trained on Caltech101 Silhouettes.

Figure 4: Dictionaries learned in temporal TruG-RBM for Bouncing Ball. (a): W1; (b): W2.

Table 1: Learned lower and upper truncation points averaged over 10 trials.

| Dataset | Lower Trun. Point | Upper Trun. Point |
|---|---|---|
| Boston Housing | -0.329 | 2.833 |
| Concrete Strength | -0.898 | 2.188 |
| Energy Efficiency | -0.528 | 1.574 |
| Kin8nm | -0.331 | 7.759 |
| Naval Propulsion | -2.755 | 4.105 |
| Cycle Power Plant | 0 | 3.131 |
| Protein Structure | -0.50 | 2.76 |
| Wine Quality Red | -0.847 | 1.725 |
| Yacht Hydrodynamic | -0.65 | 2.215 |
| Year Prediction MSD | -0.230 | 2.940 |

Figure 5: Nonlinearities learned for different datasets on the supervised TruG-TGGMs, with EE, CS and YPM being the abbreviations of Energy Efficiency, Concrete Strength, Year Prediction MSD, respectively.