[Reviews · NeurIPS 2017]

Reviewer 1



Overview: this paper introduces the Truncated Gaussian (TruG) unit (as per eq 1, and its expectation in eq 2). By adjusting the cutoffs \xi_1 and \xi_2 this can mimic ReLU and sigmoid/tanh units. It can be used as a stochastic unit in RBMs (sec 3), in temporal RBMs (sec 4), and in the TGGM (truncated Gaussian graphical model, sec 5). An old but relevant reference is "Continuous sigmoidal belief networks trained using slice sampling" B. J. Frey, in M. C. Mozer, M. I. Jordan and T. Petsche (eds), Advances in Neural Information Processing Systems 9, 452-459, January 1997. http://www.psi.toronto.edu/~psi/pubs2/1999%20and%20before/continuous_sigmoidal_belief_networks_tra_1613871.pdf One might criticise this paper by saying that once one has come up with the TruG unit, one simply has to "turn the handle" on the usual derivations to get TruG-RBMs, temporal TruG-RBMs and TruG-TGGMs. One things that I find odd is that is very well known in statistical physics that one can covert between 0/1 sigmoid coding and -1/1 tanh coding for binary spin systems by a rescaling of weights and some additive offsets. (This can often be a tutorial exercise for a Boltzmann machines class.) So I find it surprising that in Table 1 there is not more agreement between the TruG[0,1] and TruG[-1,1] results. In am also surprised that the experimental results for TruG-RBM shows such differences between the TruG[0,1] and TruG[-1,1] and the RBM results from [21] and [22], given the closeness in Figs 1(b) and 1(c). The results in table 1 with learned truncation points show modest gains over the ReLU-like TruG[0,\inf] model. For the TruG-temporal RBM we again see gains (probably larger here) over the TRBM and RTRBM baselines. Similar conclusions also apply to Table 4 for the TruG-TGGM. Overall the results feel like modest gains, but almost always with the differences in favor of TruGs. Quality: I believe the paper is technically correct (although I have not checked the math). Clarity: clearly written. Originality: one can invent Boltzmann machines with many different types of units. This feels like "adding another one", but it is quite nice how the TruG can mimic both ReLU and sigmoidal units. Significance: it is not clear that this will have very much impact, although (the deterministic version of) such units could be used in standard deep networks, allowing a smooth transition from sigmoid to ReLU units.

Reviewer 2



Summary: The paper uses doubly truncated Gaussian distributions to develop a family of stochastic non-linearities parameterized by the truncation points. In expectation, different choices of truncation points are shown to recover close approximations to popularly employed non-linearities. Unsurprisingly, learning the truncation points from data leads to improved performance. Quality and Clarity — The authors present a technically sound piece of work. The proposed family of non-linearities are sensible and the authors do a good job of demonstrating their utility across a variety of problems. The manuscript is sufficiently clear. Originality and Significance - The connection between the convolution of a Gaussian with an appropriately truncated Gaussian and popular neural network nonlinearities are well known and have been previously explored by several papers in supervised contexts (see [1, 2, 3]). In this context, learning the truncation points is a novel but obvious and incremental next step. To me, the interesting contribution of this paper lies in its empirical demonstration that learning the non-linearities leads to tangible improvements in both density estimation and predictive performance. It is also interesting that the learned non-linearities appear to be sigmoid-relu hybrids. Detailed Comments: 1) Figures 2 indicates that the learned upper truncations are all relatively small values, in the single digits. This could be a consequence of the initialization to 1 and the low learning rates, causing the recovered solutions to get stuck close to the initial values. Were experiments performed with larger initial values for the upper truncation points? 2) a) Why are the s-learn experiments missing for TruG-TGGM? b) Table 4 seems to suggest that stochasticity alone doesn’t lead to significant improvements over the ReLU MLP. Only when learning of non-linearities is added does the performance improve. What is unclear from this is whether stochasticity helps in improving performance at all, if the non-linearities are being learned. I will really like to see comparisons for this case with parameterized non-linearities previously proposed in the literature (citations 6/7/8 in the paper) to tease apart the benefits of stochasticity vs having more flexible non-linearities. Minor: There exists related older work on learning non-linearities in sigmoidal belief networks [4], which should be cited. [1] Hernández-Lobato, José Miguel, and Ryan Adams. "Probabilistic backpropagation for scalable learning of bayesian neural networks." International Conference on Machine Learning. 2015. [2] Soudry, Daniel, Itay Hubara, and Ron Meir. "Expectation backpropagation: Parameter-free training of multilayer neural networks with continuous or discrete weights." Advances in Neural Information Processing Systems. 2014. [3] Ghosh, Soumya, Francesco Maria Delle Fave, and Jonathan S. Yedidia. "Assumed Density Filtering Methods for Learning Bayesian Neural Networks." AAAI. 2016. [4] Frey, Brendan J., and Geoffrey E. Hinton. "Variational learning in nonlinear Gaussian belief networks." Neural Computation 11.1 (1999): 193-213.

Reviewer 3



This paper proposes an approach to setting an/or learning the nonlinearities in neural networks, which includes the classical sigmoid, hyperbolic tangent, and rectified linear unit (ReLU) as particular cases. The method widens the object of the learning algorithm from its linear combination weights, to include also the possibility of learning the nonlinearities. The contribution is solid and novel. Although it builds on closely related work in [31,5], I believe it extends those works enough to merit publication at NIPS.